# Rational Computational Design of Fourth-Generation EGFR Inhibitors to Combat Drug-Resistant Non-Small Cell Lung Cancer

**DOI:** 10.3390/ijms21239323

**Published:** 2020-12-07

**Authors:** Hwangseo Park, Hoi-Yun Jung, Kewon Kim, Myojeong Kim, Sungwoo Hong

**Affiliations:** 1Department of Bioscience and Biotechnology, Institute of Anticancer Medicine Development, Sejong University, 209 Neungdong-ro, Kwangjin-gu, Seoul 05006, Korea; 2Center for Catalytic Hydrocarbon Functionalizations, Institute for Basic Science (IBS), Daejeon 34141, Korea; spurs9@kaist.ac.kr (H.-Y.J.); kkw105701@kaist.ac.kr (K.K.); mymyo24@kaist.ac.kr (M.K.); 3Department of Chemistry, Korea Advanced Institute of Science and Technology (KAIST), Daejeon 34141, Korea

**Keywords:** virtual screening, de novo design, EGFR, fourth-generation inhibitor, drug resistance

## Abstract

Although the inhibitors of singly mutated epidermal growth factor receptor (EGFR) kinase are effective for the treatment of non-small cell lung cancer (NSCLC), their clinical efficacy has been limited due to the emergence of various double and triple EGFR mutants with drug resistance. It has thus become urgent to identify potent and selective inhibitors of triple mutant EGFRs resistant to first-, second-, and third-generation EGFR inhibitors. Herein, we report the discovery of potent and highly selective inhibitors of EGFR exon 19 p.E746_A750del/EGFR exon 20 p.T790M/EGFR exon 20 p.C797S (d746-750/T790M/C797S) mutant, which were derived via two-track virtual screening and de novo design. This two-track approach was performed so as to maximize and minimize the inhibitory activity against the triple mutant and the wild type, respectively. Extensive chemical modifications of the initial hit compounds led to the identification of several low-nanomolar inhibitors of the d746-750/T790M/C797S mutant. Among them, two compounds exhibited more than 10^4^-fold selectivity in the inhibition of EGFR^d746-750/T790M/C797S^ over the wild type. The formations of a hydrogen bond with the mutated residue Ser797 and the van der Waals contact with the mutated residue Met790 were found to be a common feature in the interactions between EGFR^d746-750/T790M/C797S^ and the fourth-generation inhibitors. Such an exceptionally high selectivity could also be attributed to the formation of the hydrophobic contact with a Gly loop residue or the hydrogen bond with Asp855 in the activation loop. The discovery of the potent and selective EGFR^d746-750/T790M/C797S^ inhibitors were actually made possible by virtue of the modified protein–ligand binding free energy function involving a new hydration free energy term with enhanced accuracy. The fourth-generation EGFR inhibitors found in this work are anticipated to serve as a new starting point for the discovery of anti-NSCLC medicines to overcome the problematic drug resistance.

## 1. Introduction

Epidermal growth factor receptor (EGFR) plays a key role in regulating various intracellular signaling for tumor cell proliferation, differentiation, migration, and invasion [1,2]. Deregulated activity of EGFR kinase is responsible for the pathogenesis and the progression of approximately 10–15% of lung adenocarcinomas, which are the main subtype of non-small cell lung cancer (NSCLC) [3,4,5]. The most constitutively activating mutations include the L858R single point mutation and the deletion of Glu746-Ala750 (d746-750) [6,7]. In this regard, it has served as a good therapeutic treatment of NSCLC to impair the kinase activity of the oncogenic mutants with first-generation EGFR inhibitors [8,9,10,11].

However, targeting the L858R and d746-750 mutants by the first-generation EGFR inhibitors became problematic with the manifestation of drug resistance in NSCLC patients. This acquired resistance was due to a secondary somatic mutation (T790M) in the hinge region [12,13,14,15], which caused the increase in the steric repulsions with the first-generation inhibitor drugs. This triggered the discovery of a number of the second-generation EGFR inhibitors to surmount the drug resistance [16]. Despite the alleviation of drug resistance, the clinical application of the second-generation EGFR inhibitors was limited owing to the severe side effects caused by the simultaneous inhibition of the mutant and the wild-type EGFR. Accordingly, a great deal of efforts has been devoted to the discovery of the new EGFR inhibitors specific for the drug-resistant d746-750/T790M and L8585R/T790M double mutants [17,18,19], which was characteristic of the third-generation inhibitors.

A common mechanistic feature of the second- and the third-generation EGFR inhibitors lies in the irreversible inhibition of the mutant kinase activity through the formation of a covalent bond with the nucleophilic side chain of Cys797. This is made possible because the inhibitors possess a suitable electrophilic group susceptible to nucleophilic attack by the side-chain thiolate group of Cys797 at the lip of the ATP-binding site. However, a drug resistance problem was raised again with the appearance of a tertiary mutation (C797S), which has the effect of preventing the covalent interactions with residue 797 [20,21,22,23]. The identification of L858R/T790M/C797S and d746-750/T790M/C797S mutant inhibitors has therefore been actively pursued to target the NSCLC cells resistant to the second- and the third-generation EGFR inhibitors. Although most triple mutant inhibitors were insufficient as a lead compound due to the higher biochemical potency against the wild type [24,25,26], some derivatives of 2-aryl-4-aminoquinazoline and 5-methylpyrimidopyridone proved to be good candidates for anti-NSCLC agent with high selectivity in the inhibition of the triple mutant [27,28]. The triple mutant inhibitors to suppress the resistance induced by C797S mutation are termed the fourth-generation EGFR inhibitors for which selectivity over the wild-type EGFR is also necessary to circumvent the severe potential side effects [29]. In recent years, the discovery of the fourth-generation inhibitors to specifically inhibit the triple EGFR mutants responsible for the resistance to second- and third-generation inhibitor drugs has become the focus of anti-NSCLC drug development [30].

This work aimed at the discovery of the potent and specific inhibitors of the d746-750/T790M/C797S mutant EGFR based on the two-track virtual screening, de novo design, chemical synthesis, and enzyme inhibition assays. The two-track approach was adopted to find molecules with maximal and minimal inhibitory activity against the triple mutant and the wild type, respectively. To augment the likelihood of finding the new fourth-generation EGFR inhibitors in the early stage of discovery, the protein–ligand binding free energy function to evaluate the putative inhibitors was modified by introducing an adequate hydration energy function. This preliminary step is necessary because a poor correlation between the computational and experimental results for protein–ligand binding affinity stems in a large part from the underestimation of ligand hydration effects [31]. Through the binding mode analysis with docking simulations, we also address the energetic and structural features associated with the mechanism of action for the fourth-generation EGFR inhibitors.

## 2. Results and Discussion

### 2.1. Structure-Based Virtual Screening of the Fourth-Generation EGFR Inhibitors

Rational design of the fourth-generation EGFR inhibitors was a formidable task because of the lack of 3D structure of the d746-750/T790M/C797S triple mutant. Accordingly, the virtual screening and de novo design started with the preparation of a relevant structural model for the kinase domain of the d746-750/T790M/C797S mutant by homology modeling. The atomic coordinates of the target protein were produced using the active form of the L858R/T790M/C797S mutant as the structural template [28]. The optimized structural model for the d746-750/T790M/C797S mutant was validated with the ProSa 2003 program, which has been widely used as a valuable computational tool for examining whether the interactions of each amino-acid residue with the rest part of the protein are maintained favorably [32]. This free energy profile for individual amino acids could be obtained with the knowledge-based mean field potential.

Figure 1 shows the free energy profile of the homology-modeled kinase domain of the d746-750/T790M/C797S mutant in comparison with that of the L858R/T790M/C797S mutant, which served as the structural template. It is remarkable to see that the target protein exhibits the better energy profile than the template in most parts of the protein especially in the central region between the N- and C-terminal domains. Furthermore, the energy values remain negative throughout the amino acid residues, implying that the homology-modeled structure of the d746-750/T790M/C797S mutant would be physically acceptable. Based on the good energetic features, the structure of the kinase domain of the d746-750/T790M/C797S mutant constructed with homology modeling was adopted as the receptor model for the virtual screening and de novo design to find the fourth-generation EGFR inhibitors.

The applicability of virtual screening and de novo design has been limited owing to the inaccurate nature of the scoring function for estimating the protein–ligand binding affinity [33,34]. In particular, the binding free energy functions of popular docking programs tend to underestimate the ligand dehydration effects in the protein–ligand association. This may inevitably culminate in overestimating the biochemical potency of a molecule with many hydrophilic groups [31]. Prior to conducting the virtual screening, therefore, we modified the dehydration term in the original scoring function within the framework of the extended solvent-contact model to enhance the predictive capability.

Virtual screening began with the preparation of a chemical library from which the tight-binding inhibitor scaffolds for the d746-750/T790M/C797S mutant could be found by docking simulations. This docking library was constructed by collecting a total of approximately 370,000 ‘Rule-of-Five’-compliant molecules [35] with molecular weights ranging from 300 and 400 atomic mass units (amu) among the commercially available compounds. Virtual screening of the fourth-generation EGFR inhibitors was then carried out through the docking simulations in the ATP-binding pockets of the d746-750/T790M/C797S mutant and the wild type of EGFR. This two-track approach was intended to collect the putative inhibitors with binding free energies lower than -10 kcal/mol and higher than −6.0 kcal/mol with respect to the triple mutant and the wild type, respectively, which corresponds to at least 1000-fold difference in binding affinities. As a result that the bidentate hydrogen-bond interactions in the hinge region were characteristic of the effective fourth-generation EGFR inhibitors [27,28], only molecules capable of forming the two hydrogen bonds with backbone groups of residues 791-795 were selected with the distance criteria of <3.5 Å after all molecules in the docking library were screened.

Only 26 compounds satisfied all the filtration criteria as indicated in Figure 2. This small number of virtual hits exemplifies the difficulty in discovering the fourth-generation inhibitors that are specific for the triple mutant. This can be understood in the context of the high structural similarity between the mutant and the wild type. All virtual hits were assessed for the presence of inhibitory activity against the d746-750/T790M/C797S mutant and the wild type via in vitro radiometric ([γ-^33^P]-ATP) kinase assays (Reaction Biology Corp., Malvern, PA, USA). As a consequence of virtual and experimental screening, four molecules were identified as low-micromolar inhibitors of d746-750/T790M/C797S mutant. Actually, none of the four molecules have previously been reported as EGFR kinase inhibitors. With respect to disproving the possibility of acting as a false positive in enzyme inhibition assays, all four inhibitors were examined in the public ZINC database [36] to confirm the lack of any substructure in pan assay interference compounds (PAINS) [37].

Chemical structures and biochemical potencies of the four inhibitors of d746-750/T790M/C797S mutant EGFR are summarized in Figure 3 and Table 1, respectively. It is a common structural feature for **1**–**4** to possess several polar groups and nonpolar aromatic rings, therefore, both hydrogen-bond and van der Waals interactions are likely to serve as the major binding forces to accommodate them in the ATP-binding site of the d746-750/T790M/C797S mutant.

As listed in Table 1, all four inhibitors exhibited good biochemical potency against the triple mutant with IC_50_ values ranging from 0.8 to 3 μM. Furthermore, compounds **1–3** appear to be selective in the inhibition of the triple mutant over the wild type by a factor of 10–35. They are therefore worthy of further development with structural modifications to maximize the inhibitory activity against NSCLC cells resistant to second- and third-generation EGFR inhibitor drugs. **1** seems to serve as an effective molecular core for designing the new fourth-generation EGFR inhibitors due to the highest selectivity and good inhibitory activity against the triple mutant. **2** and **3** contain 2-aryl-4-aminoquinazoline and 1,3,5-triazin-2-amine moieties, respectively, which were also included in the previously identified inhibitors of the triple mutant [27].

It is quite unexpected that **4** reveals 136-fold higher inhibitory activity against the wild type than against the d746-750/T790M/C797S mutant, indicating that its binding affinity for the wild type was underestimated to a great extent in the precedent virtual screening. Thus, the scoring function used in this work remains imperfect despite the modification by substituting a sophisticated hydration energy term. Although **4** was found accidentally as a potent inhibitor of the wild-type EGFR, it deserves consideration for further development as a new first-generation inhibitor because the biochemical potency reached the nanomolar level (Table 1) without any chemical modification.

To find the structural relevance for low-micromolar activity of the newly discovered EGFR inhibitors, their binding modes were derived with docking simulations using the modified scoring function. Overlaid in Figure 4 are the docked poses of **1–4** around the ATP-binding site of the d746-750/T790M/C797S mutant. All four inhibitors appear to be accommodated in the well-established binding pocket comprising the glycine-rich loop (Gly loop, residues 718–726), the hinge region (residues 791–795) of the ATP-binding site, and the residues at the interface of the N- (700–860) and C-terminal (861–1014) domains. It is also a common complexation pattern that the inhibitors reside in close proximity to the side chains of Met790 and Ser797, which were mutated from Thr790 and Cys797 of the wild-type EGFR, respectively. The interactions with these mutated residues would be necessary for the specific inhibition of the d746-750/T790M/C797S mutant. To find the potential allosteric sites that could accommodate **1–4**, additional docking simulations were carried out using 3D grid maps extended to include the whole EGFR kinase domain. Nonetheless, no peripheral binding site was identified in which **1–4** could be stabilized with a negative value of binding free energy. Therefore, compounds **1–4** are likely to impair the kinase activity of the d746-750/T790M/C797S mutant through the specific binding in the ATP binding site.

As a result that **1** and **2** are low-micromolar inhibitors of the d746-750/T790M/C797S mutant with reasonably good selectivity over the wild type, their complexation patterns in the ATP-binding site would provide great insight into how to promote the potency and selectivity through chemical modifications. Figure 5 illustrates the most stable binding modes **1** and **2** obtained with docking simulations using the modified scoring function. The carbonyl oxygen of **1** receives a hydrogen bond from the backbone amidic moiety of Met793 in the hinge region while the phenolic moiety donates a hydrogen bond to the backbone aminocarbonyl oxygen of Leu718 in the Gly loop. These two hydrogen bonds seem to play a key role in binding of **1** to the mutant EGFR because they are established in the middle of the ATP-binding pocket. The third hydrogen bond is observed between the terminal piperidine moiety of **1** and the sidechain hydroxyl group of Ser797, which augments the strength of EGFR^d746-750/T790M/C797S^-**1** binding. This interaction would also be important in terms of selectivity because it involves the mutated sidechain at residue 797. **1** looks stabilized further through the hydrophobic contacts with the nonpolar side chains of Leu718, Val726, Phe723, Met790, Leu792, and Leu844. In particular, we note that three leucine residues (Leu718, Leu792, and Leu844) accommodate the 6-hydroxybenzofuran-3-one moiety of **1** around the two hydrogen bonds in the ATP-binding site. These interactions are assumed to contribute to the strengthening of the vicinal hydrogen bonds by preventing the approach of solvent water molecules. Significant synergistic effects are therefore anticipated in binding affinity by positioning the two hydrogen bonds in proximity to the hydrophobic contacts. Indeed, it has served as a facile strategy in the optimization of biochemical potency to reinforce the hydrogen bonds cooperatively with the hydrophobic interactions in complexation with the target proteins [38,39].

The binding mode of **2** in the ATP-binding site of EGFR^d746-750/T790M/C797S^ is similar to that of **1** in that three hydrogen bonds are involved in the complexation. In particular, we note that one of the nitrogens on the pyrimidine ring and the terminal phenolic moiety of **2** receives and donates a hydrogen bond from the backbone amidic nitrogen of Met793 and to the aminocarbonyl oxygen of Gln791, respectively. These bidentate hydrogen bonds are supported by the neighboring van der Waals contacts between the inhibitor phenyl ring and the sidechain of the mutated Met790. This kind of interaction pattern was actually observed in a variety of EGFR-inhibitor complexes, irrespective of the mutational status of EGFR [40]. In the calculated structure of EGFR^d746-750/T790M/C797S^-**2** complex, an additional hydrogen bond is found at the bottom of the ATP-binding site between the –NH moiety attached to the pyrimidine ring of **2** and the side-chain hydroxyl group of the mutated Ser797. This hydrogen bond seems to contribute to the selective inhibition of the triple mutant because the substitution of serine for cysteine at residue 797 is the tertiary mutation to afford the resistance to the second- and third-generation EGFR inhibitors. The stabilization of **2** in the ATP-binding pocket may be promoted due to the hydrophobic interactions with the nonpolar side chains of Leu718, Val726, Phe723, Met790, Leu792, and Leu844. On the basis of the structural features in the calculated EGFR^d746-750/T790M/C797S^-**2** complex, it is most likely that the low-micromolar inhibitory activity of **2** would stem from the multiple hydrogen bonds facilitated by the hydrophobic contacts in the ATP-binding site.

As a result that **1** is almost fully accommodated in the ATP-binding site of the d746-750/T790M/C797S mutant, only small substituents would be allowed for derivatizations to optimize the inhibitory activity. On the other hand, the terminal phenolic moiety of **2** is directed to a vacant peripheral binding pocket consisting of Val726, Lys745, Met790, Thr854, and Asp855 (Figure 5). This sub-binding region may serve as a potential target for improving both the potency and selectivity in the inhibition of deregulated EGFR mutants because it includes the amino-acid residues not only in the hinge region (Met790) and Gly loop (Val726) but also in the DFG motif (Asp855-Phe856-Gly857) that resides on the activation loop. Therefore, the introduction of a suitable chemical moiety at the terminal phenolic moiety of **2** would have the effect of enhancing the biochemical potency against the triple mutant.

### 2.2. Synthesis of the Derivatives of 1 and 2 Generated from de novo Design

Virtual screening and the subsequent de novo design processes generated a variety of fourth-generation EGFR inhibitors with **1** and **2** as the molecular core. Aurone derivatives **C** were synthesized as illustrated in Scheme 1. The key intermediates **B** were prepared by Mannich reaction of benzofuran-3(2H)-one **A** with paraformaldehyde and various secondary amines. Subsequently, base-catalyzed condensation of **B** with aldehyde gave rise to the desired aurone derivatives **C**.

Quinazoline derivatives **E** were prepared as outlined in Scheme 2. Base-promoted nucleophilic addition of requisite amines to 2,4-dichloroquinazoline **C** led to the installation of the amino groups at the C4 position of quinazolines **D**. Next, Suzuki−Miyaura coupling reactions between intermediate **D** with various arylboronic acids or arylboronic acid pinacol esters yielded target compound **E**.

### 2.3. Biochemical Potencies of the Newly Synthesized Compounds

Most initial hit compounds derived from virtual and experimental screening tend to be insufficient in both biochemical potency and selectivity to serve as a good lead compound because all the candidate molecules were prepared for the purposes other than inhibiting the target protein of interest. Hence, we performed de novo design to identify the fourth-generation EGFR inhibitors better than **1** using this compound as the molecular core. Based on the calculated binding mode of **1** (Figure 5), the derivatizations were conducted so as to maximize the interactions with Ser797 and the amino-acid residues in the peripheral binding pocket. Table 2 lists the chemical structures and IC_50_ data of the seven derivatives of **1** with respect to the d746-750/T790M/C797S mutant and the wild type of EGFR. Biochemical potencies for both the wild type and the triple mutant appear to remain almost intact as the methyl group on the terminal piperidine ring of **1** is removed in **5** or shifted to the neighboring carbons in **6** and **7**. Further substitution of the methoxy moiety in **8** causes a substantial loss of inhibitory activity, confirming that a bulky substituent on the benzofuran ring would be disfavored for tight binding to EGFR. In contrast to the negligible substitution effects on the terminal piperidine ring, the replacement of piperidine in **5** with piperazine in **9** leads to the increase in the inhibitory activity by a factor of 3 with respect to both target proteins. Although a negative effect of the methoxy substitution is also observed in **10**, the biochemical potency against the triple mutant increases further to the submicromolar level due to the introduction of -OH moiety in **11** on the benzofuran ring of **9**. However, the loss of selectivity over the wild-type EGFR makes it difficult for **11** to serve as a good lead for the development of a new anti-NSCLC medicine.

As a result that it was unsuccessful to derive the effective fourth-generation EGFR inhibitors by the derivatizations of **1**, the alternative de novo designs were carried out using **2** as a new molecular core. Based on the structural features derived from docking simulations between the d746-750/T790M/C797S mutant and **2**, the structural modifications aimed at finding the optimal chemical moiety (R1) attached to the nitrogen of the central quinazolin-4-amine group. The *para* position (R2) of the terminal phenol group was also selected as the substitution point because its importance in EGFR inhibition was demonstrated in the previous study [27]. Actually, only small substituents are allowed at the R2 position because the terminal phenyl ring of **2** resides in close proximity to the side chains of Val726, Met790, and Thr854 in the ATP-binding site (Figure 5). Among the ten derivatives of **2** designed as new fourth-generation EGFR inhibitors, two candidates exhibited not only the low nanomolar activity but also high selectivity in the inhibition of the triple mutant over the wild type by a factor of more than 10^4^.

Summarized in Table 3 are the IC_50_ values of various derivatives of **2** generated from de novo design with respect to the d746-750/T790M/C797S mutant and the wild type of EGFR. All derivatives reveal the higher biochemical potency against the triple mutant than against the wild type. This exemplifies the usefulness of the modified scoring function adopted in this work for optimizing the activities of the fourth-generation EGFR inhibitors as well as for identifying the molecular cores of potent inhibitors in virtual screening. Although the introduction of *N*,*N*-diethylpropan-1-amine (**12**) and 4-methylthiazole (**14**) moiety at the R1 position leads to only a slight increase in the inhibitory activity against the triple mutant, the biochemical potency increases to the submicromolar level by further substitution of a nitrile group at the R2 position in **13** and **15**. The submicromolar inhibitory activity is retained in the presence of 3-methylpyrrole (**16**) and 4-methyloxazole (**17**) at the R1 position. Most remarkably, the replacement of the five-membered aromatic ring with imidazole (**18**) lead to the increase in the biochemical potency to low-nanomolar level with more than 6690-fold higher activity against the d746-750/T790M/C797S mutant than against the wild-type EGFR. The selectivity index surges to higher than 10^4^ either by the methyl substitution at the nitrogen atom of the imidazole ring (**19**) or by the one-carbon elongation of the linking group in **20** between the central quinazoline and the terminal imidazole ring. The change of R2 substituent from nitrile in **19** to aldehyde in **21** leads to a substantial loss of selectivity in contrast to the maintenance of low-nanomolar inhibitory activity against the triple mutant, which confirms the necessity of the former to retain both biochemical potency and selectivity. Among the ten derivatives of **2** listed in Table 3, **18**–**20** may be proposed as new promising fourth-generation EGFR inhibitors in the context of the exceptionally high selectivity and low-nanomolar activity in the inhibition of the d746-750/T790M/C797S mutant. They are anticipated to serve as a good starting point for the development of new medicines against NSCLC cells with acquired resistance to second- and third-generation EGFR inhibitor drugs.

To find a rationale for the exceptionally high selectivity and biochemical potency, the binding modes of **19** and **20** were explored with docking simulations in the ATP-binding site of the d746-750/T790M/C797S mutant. The calculated binding configurations of the two nanomolar inhibitors are compared in Figure 6. Both **19** and **20** are supposed to be well accommodated in the ATP-binding site in the similar way to **2** (Figure 5) with the nitrile moiety occupying the peripheral binding pocket. This additional interaction would contribute to the increase in selectivity as well as in the inhibitory activity in going from **2** to **19** and **20** (Table 3) because the peripheral binding pocket involves the mutated residue Met790.

It is interesting to note that the terminal 1-methylimidazole group of **19** forms a close van der Waals contact with the sidechain phenyl ring of Phe723 in the Gly loop. This interaction would also be a significant contributor to the impairment of the kinase activity of the EGFR^d746-750/T790M/C797S^ because the Gly loop acts as a receptor for binding of the phosphate group in complexation with ATP. The extra stabilization of the EGFR^d746-750/T790M/C797S^-**20** complex seems to be caused by the additional hydrogen-bond interaction of the terminal imidazole moiety with the side-chain carboxylate group of Asp855, which is a component of the DFG motif (Asp855-Phe856-Gly857) on the activation loop. This new hydrogen bond is made possible by one-carbon elongation of the linking group to connect the central quinazoline ring and the terminal imidazole group, which is required to reach Asp855 that resides at the end of the ATP-binding pocket. In the calculated EGFR^d746-750/T790M/C797S^-**19** and EGFR^d746-750/T790M/C797S^-**20** complexes, it is also worth noting that both the hydrophobic interaction of **19** with Phe723 and the hydrogen bond of **20** with Asp855 are supported by the neighboring hydrogen bond with the side-chain hydroxyl moiety of mutated residue Ser797. The additional interactions generated by the chemical modification from **2** to **19** and **20** are thus consistent with the substantial increase in the inhibitory activity and selectivity.

Two promising molecules (**18** and **19**) were further evaluated for cellular growth inhibition against Ba/F3 cell lines with the wild type and the d746-750/T790M/C797S mutant EGFR. These cell-based assays were conducted at WuXi AppTec Corp. (Shanghai, China) using Gefitinib and Brigatinib as the reference compounds, which were approved by FDA for the NSCLC therapy. Only the two molecules were investigated because of the difficulty in synthesizing **20** and **21** in the amount sufficient to perform cellular studies. As shown in Table 4, both **18** and **19** revealed high antiproliferative activity against the d746-750/T790M/C797S mutant cell at the submicromolar level. However, the micromolar-level inhibitory activity was also observed for the cell lines with the wild-type EGFR to the extent similar to the reference compounds. As a result that the anticellular activity appears to be higher than the inhibitory activity in enzyme assays (Table 3), further optimization would be required for the fourth-generation EGFR inhibitors found in this work to alleviate the potential off-target activity.

Although some promising fourth-generation EGFR inhibitors were identified in this work, the biochemical potencies of many compounds in Table 1, Table 2 and Table 3 remained modest in spite of the modification of the scoring function for virtual screening and de novo design. The imperfection of the scoring function can be attributed in a large part to the incomplete optimization of the weighting factors for varying energy terms, which stems from the insufficient number of EGFR-inhibitor complexes in the training set for parameterizations. The scoring function is expected to become even more accurate by reoptimizing the weighting factors using the new training set supplemented with a variety of EGFR-inhibitor complexes. In the near future, we plan to design and identify the more potent and selective fourth-generation EGFR inhibitors than those presented in this work with the improved scoring function.

## 3. Materials and Methods

### 3.1. Structural Preparations of d746-750/T790M/C797S Mutant and Wild Type of EGFR

As a result that the 3D structure of d746-750/T790M/C797S mutant EGFR was unavailable in Protein Data Bank (PDB), its atomic coordinates were constructed by homology modeling using the active conformation of the L858R/T790M/C797S mutant [28] as the structural template (PDB entry: 6JRJ). This homology modeling began with the retrieval of the amino acid sequence of human EGFR comprising 1210 residues from UniProtKB protein knowledgebase (http://www.uniprot.org, accession number: P00533, gene name: *EGFR ERBB*). Only the cytoplasmic kinase domain (residues 700-1014) of EGFR was considered in the homology modeling because the present study was focused on the discovery of ATP-competitive inhibitors. To build the structure of the d746-750/T790M/C797S mutant, the atomic coordinates were optimized in such a way as to minimize the violation of spatial restraints as implemented in the latest version of the MODELLER program [41].

The X-ray crystal structure of the EGFR kinase domain in the active form [42] (PDB entry: 2GS2) was selected as the receptor model for the wild type. Finally, all-atom models of the wild type and d746-750/T790M/C797S mutant were constructed by adding the hydrogen atoms according to the protonation states of titratable residues revealed in the patterns of intramolecular hydrogen bonds.

### 3.2. Two-Track Virtual Screening to Identify the Fourth-Generation EGFR Inhibitors

To identify the EGFR inhibitors specific for the d746-750/T790M/C797S mutant, virtual screening should be conducted not only for the triple mutant but also for the wild-type EGFR to collect the candidates capable of binding tightly and weakly to the former and the latter, respectively. This two-track virtual screening started with the preparation of a docking library containing approximately 370,000 synthetic and natural compounds from the latest version of the chemical database provided by InterBioScreen Ltd. A total of 560,000 compounds in the database were filtered to collect only those that possess the physicochemical properties of drug candidates [35]. Similar molecules with the Tanimoto coefficient exceeding 0.8 were then clustered into a single representative molecule to remove structural redundancy. After the two filtration steps, 3D atomic coordinates of all the molecules in the docking library were generated with the CORINA program [43].

Two-track virtual screening to identify fourth-generation EGFR inhibitors was performed with the automated AutoDock program [44,45], the performance of which had been appreciated in the discovery of various kinase inhibitors [46,47,48]. Despite the significant contribution to protein–ligand binding affinity, it is difficult to precisely reflect the ligand hydration effects in docking simulations for virtual screening because the scoring function of the original AutoDock program encompasses a crude dehydration energy term involving only six atom types to describe varying solute molecules. As a preliminary step to virtual screening of fourth-generation EGFR inhibitors, the scoring function was modified by substituting a new ligand dehydration energy term for the original one. This modified scoring function (∆*G_b_^aq^*) has the following mathematical form.
(1)$$\begin{array}{l} \Delta G_{b}^{aq}=W_{vdW}\sum_{i=1}\sum_{j=1}\left(\frac{A_{ij}}{r_{ij}^{12}}-\frac{B_{ij}}{r_{ij}^{6}}\right)+W_{hbond}\sum_{i=1}\sum_{j=1}E(t)\left(\frac{C_{ij}}{r_{ij}^{12}}-\frac{D_{ij}}{r_{ij}^{10}}\right)\\ \;\;\;\;+W_{elec}\sum_{i=1}\sum_{j=1}\frac{q_{i}q_{j}}{\varepsilon (r_{ij})r_{ij}}+W_{tor}N_{tor}+\sum_{i=1}S_{i}\left(O_{i}^{\max}-\sum_{j\neq i}V_{j}e^{-\frac{r_{ij}^{2}}{2\sigma^{2}}}\right) \end{array}$$

The coefficients *W_vdW_*, *W_hbond_*, *W_elec_*, and *W_tor_* in Equation (1) refer to the weighting factors of van der Waals interactions, hydrogen bonds, electrostatic interactions, and torsional motions of the putative inhibitor, respectively. The variable *r_ij_* represents the interatomic separation, and the *A_ij_*, *B**_ij_*, *C**_ij_*, and *D_ij_* parameters are associated with the well depth and the equilibrium distance in a given potential energy function. AMBER force field parameters were adopted to compute the van der Waals interaction energies between EGFR and all putative inhibitors. An additional weighting factor (*E*(*t*)) was necessary in the intermolecular hydrogen bond term to reflect the angle-dependent directionality. In calculating the intermolecular electrostatic interactions between EGFR and a putative inhibitor, we used the atomic charges determined with Gasteiger–Marsilli method [49] and the distance-dependent sigmoidal function as the dielectric constant to simulate the long-range charge screening effects [50]. The *N_tor_* parameter in the torsional term means the number of rotatable bonds to estimate the entropic penalty for a putative inhibitor to be bound in the ATP-binding site of EGFR.

The first four terms in Equation (1) correspond to the binding free energy in the gas phase, while the final term is the negative of ligand hydration free energy. In this energy term, the *S_i_*, *V_i_*, and *O_i_*^max^ parameters denote the atomic hydration energy per unit volume, the atomic volume in molecules, and the atomic maximum occupancy, respectively [51]. To quantify the hydration free energies of putative EGFR inhibitors, all atomic parameters were derived with the extended solvent-contact model that had achieved high outperformance in the SAMPL4 blind prediction challenge for molecular hydration free energies [52,53]. The introduction of this sophisticated hydration free energy term in the scoring function would enhance the possibility of finding actual EGFR inhibitors in virtual screening by preventing the overestimation of the biochemical potency of a candidate molecule with many polar groups [31]. Actually, the virtual screening protocol was not validated explicitly in this work because the accuracy of the modified scoring function was demonstrated well in the previous studies [47,48].

### 3.3. De novo Design

To improve the potency and selectivity in inhibiting the d746-750/T790M/C797S mutant, the hit compounds identified from virtual screening needed to be structurally modified in such a way as to maximize the interactions in the ATP-binding site. For this purpose, the structure-based de novo design was performed in a stepwise fashion. First, various derivatives of a hit compound were generated with the LigBuilder program [54] using the structure of d746-750/T790M/C797S mutant in complex with the hit compound as the starting structure. This step proceeded with the genetic algorithm to change the structure of the molecular core by introducing a variety of substituents at the specified positions. The number of such substitution positions was limited to two in this study to reduce the computational burden. The empirical scoring function comprising electrostatic, van der Waals, hydrogen bond, and entropic terms was then used for selecting approximately 14,000 derivatives that were predicted to have a higher binding affinity than the initial hit. Bioavailability rules were also applied in this step to collect only the derivatives with druggable physicochemical properties.

The second step of de novo design was performed in a similar manner to the precedent two-track virtual screening in the context that all of the derivatives were further screened to select only those with a higher binding affinity for the d746-750/T790M/C797S mutant than for the wild type. The modified scoring function in Equation (1) served to evaluate the derivatives designed in the first step. Among the derivatives predicted to bind more tightly to the triple mutant than the wild type, those with the difference in binding free energies larger than 5 kcal/mol were inspected for the availability of chemical synthesis. Finally, seven and ten derivatives of **1** and **2** were synthesized and evaluated, respectively, with enzyme inhibition assays to find the new fourth-generation EGFR inhibitors.

### 3.4. Chemical Synthesis

#### 3.4.1. General Methods

Unless stated otherwise, reactions were performed in flame-dried glassware. Analytical thin layer chromatography (TLC) was performed on precoated silica gel 60 F^254^ plates and visualization on TLC was achieved by UV light (254 and 365 nm). Flash column chromatography was performed on silica gel (400–630 mesh) or a Combi*Flash*^®^
*R_f_*^+^ system with Redi*Sep*^®^
*R_f_* silica columns (230–400 mesh) using a proper eluent. ^1^H NMR was recorded on Brucker Avance 400 MHz, Brucker Avance Neo 500 MHz, or Agilent Technologies DD2 600 MHz. Chemical shifts were quoted in parts per million (ppm) referenced to the appropriate solvent peak or 0.0 ppm for tetramethylsilane. The following abbreviations were used to describe peak splitting patterns when appropriate: br = broad, s = singlet, d = doublet, t = triplet, q = quartet, p = pentet, m = multiplet, dd = doublet of doublet, td = triplet of doublet, ddd = doublet of doublet of doublet. Coupling constants, *J*, were reported in hertz unit (Hz). ^13^C NMR was recorded on Brucker Avance 100 MHz or Agilent Technologies DD2 150 MHz, and fully decoupled by broad band proton decoupling. Chemical shifts were reported in ppm referenced to the centerline of a triplet at 77.16 ppm of CDCl_3_-*d* septet at 39.52 ppm of DMSO-*d*_6_, or quintet at 53.84 of CD_2_Cl_2_-*d_2._* High-resolution mass spectra were obtained by using EI or FAB method from Korea Basic Science Institute (Daegu). Commercial grade reagents and solvents were used without further purification except as indicated below.

#### 3.4.2. Synthesis of Compound C

Generally, aurone derivatives were synthesized via two steps as depicted in Scheme 1. At the first step, piperidine derivatives or Boc protected piperazine were introduced to a coumaranone core with methylene unit using paraformaldehyde. To the corresponding modified coumaranone core, benzofuran-2-carbaldehyde was introduced by condensation reaction under catalytic amount of base. Further deprotection processes were employed to afford more aurone derivatives using HCl or BBr_3_.

#### 3.4.3. Representative Procedure for Modification of Coumaranone (Step 1)

Reaction was conducted in a round bottom flask (15 mL) sealed with a rubber septa. 6-Hydroxy-3-coumaranone (100 mg, 0.67 mmol) and paraformaldehyde (20 mg, 0.67 mmol) were combined. To the mixture, 3 mL ethanol and 3-Methylpiperidine (78 mg, 0.67 mmol) were added. The mixture was heated and stirred under reflux at 80 °C for overnight. The reaction mixture was monitored by TLC using 95% dichloromethane and 5% methanol as the mobile phase. After 24 h, the reaction mixture was concentrated and diluted with dichloromethane (25 mL × 3) and washed with brine (50 mL). The organic layer was dried over Na_2_SO_4_. After removal of solvent, concentrated mixture was purified by flash chromatography on silica gel (dichloromethane/methanol = 20:1) to give the desired product (35 mg, 20%, white solid).

#### 3.4.4. Representative Procedure for Preparing Aurone Derivatives (Step 2)

Reaction was conducted in a round bottom flask (15 mL) sealed with a rubber septa. 6-Hydroxy-7-((3-methylpiperidin-1-yl)methyl)benzofuran-3(2H)-one (35 mg, 0.13 mmol) was dissolved in 3 mL methanol. Benzofuran-2-carbaldehyde (19 mg, 0.13 mmol) and catalytic amount piperidine (1.1 mg, 0.013 mmol) were added to the solution. The mixture was heated and stirred for 2 h at 60 °C. The reaction mixture was monitored by TLC using 95% dichloromethane and 5% methanol as the mobile phase. After 2 h, the reaction mixture was concentrated under reduced vacuum, and purified by flash chromatography on silica gel (dichloromethane/methanol = 20:1) to give the desired product compound (35 mg, 68%, yellow solid).

**(Z)-2-(benzofuran-2-ylmethylene)-6-hydroxy-7-(piperidin-1-ylmethyl)benzofuran-3(2H)-one (5).** Prepared according to general procedure started with 0.67 mmol 6-hydroxy-3-coumaranon to afford the title compound (38 mg, 23%, yellow oil for step 1 using piperidine, and 12 mg, 14%, yellow solid for step 2 started with 0.23 mmol modified coumaranone derived from step 1 and benzofuran-2-carbaldehyde); ^1^H NMR (400 MHz, Chloroform-*d*) δ 7.69 (dd, *J* = 7.8, 1.2 Hz, 1H), 7.66 (d, *J* = 8.5 Hz, 1H), 7.55 (dd, *J* = 8.4, 1.0 Hz, 1H), 7.43–7.38 (m, 1H), 7.33–7.29 (m, 2H), 6.90 (s, 1H), 6.76 (d, *J* = 8.5 Hz, 1H), 4.13 (s, 2H), 2.96 (br, 6H), 1.82 (br, 4H); ^13^C NMR (100 MHz, Chloroform-*d*) δ 181.7, 168.7, 165.6, 155.4, 150.6, 148.4, 128.8, 126.0, 125.2, 123.4, 121.5, 114.1, 113.0, 111.5, 111.5, 104.1, 99.8, 54.3, 54.0, 25.6, 23.6; HRMS (EI) *m*/*z* calcd. for C_23_H_21_NO_4_^+^ [M]^+^: 375.1471, found: 375.1473.

**(Z)-2-(benzofuran-2-ylmethylene)-6-hydroxy-7-((3-methylpiperidin-1-yl)methyl)benzofuran-3(2H)-one (6).** Prepared according to general procedure started with 0.67 mmol 6-hydroxy-3-coumaranon to afford the title compound (35 mg, 20%, white solid for step 1 using 3-methylpiperidine, and 35 mg, 68%, yellow solid for step 2 started with 0.15 mmol modified coumaranone derived from step 1 and benzofuran-2-carbaldehyde); ^1^H NMR (500 MHz, Chloroform-*d*) δ 7.67–7.64 (m, 1H), 7.60 (d, *J* = 8.5 Hz, 1H), 7.52 (d, *J* = 8.2 Hz, 1H), 7.36 (ddd, *J* = 8.2, 7.1, 1.3 Hz, 1H), 7.28 (d, *J* = 7.2 Hz, 2H), 6.86 (s, 1H), 6.66 (d, *J* = 8.4 Hz, 1H), 4.04 (s, 2H), 3.19–2.98 (m, 2H), 2.30 (br, 1H), 2.01 (br, 1H), 1.88–1.77 (m, 3H), 1.76–1.66 (m, 1H), 1.04 (s, 1H), 0.94 (d, *J* = 6.4 Hz, 3H); ^13^C NMR (125 MHz, Chloroform-*d*) δ 181.9, 168.6, 165.8, 155.6, 150.7, 148.5, 128.9, 126.1, 125.4, 123.5, 121.7, 114.1, 113.3, 111.7, 111.6, 104.2, 100.0, 77.4, 61.0, 54.0, 53.6, 32.3, 25.4, 19.5; HRMS (EI) *m*/*z* calcd. for C_24_H_23_NO_4_^+^ [M]^+^: 389.1627, found: 389.1626.

**(Z)-2-(benzofuran-2-ylmethylene)-6-hydroxy-7-((4-methylpiperidin-1-yl)methyl)benzofuran-3(2H)-one (7).** Prepared according to general procedure started with 0.67 mmol 6-hydroxy-3-coumaranon to afford the title compound (17 mg, 10%, red oil for step 1 using 4-methylpiperidine, and 19 mg, 73%, yellow solid for step 2 started with 0.065 mmol modified couramanone derived from step 1 and 5-methoxybenzofuran-2-carbaldehyde); ^1^H NMR (400 MHz, Chloroform-*d*) δ 7.67 (d, *J* = 7.8 Hz, 1H), 7.60 (d, *J* = 8.4 Hz, 1H), 7.53 (d, *J* = 8.2 Hz, 1H), 7.41–7.33 (m, 1H), 7.28 (q, *J* = 4.1, 3.1 Hz, 2H), 6.87 (s, 1H), 6.65 (d, *J* = 8.5 Hz, 1H), 4.05 (s, 2H), 3.23–3.02 (m, 2H), 2.47–2.25 (m, 2H), 1.87–1.73 (m, 2H), 1.55 (br, 1H), 1.45–1.31 (m, 2H), 1.00 (d, *J* = 6.4 Hz, 3H); ^13^C NMR (101 MHz, Chloroform-*d*) δ 181.8, 168.6, 165.7, 155.5, 150.7, 148.5, 128.9, 126.1, 125.3, 123.5, 121.7, 114.1, 113.2, 111.6, 111.6, 104.3, 99.9, 53.9, 53.7, 34.0, 30.4, 21.7; HRMS (FAB) *m*/*z* calcd. for C_24_H_24_NO_4_^+^ [M]^+^: 390.1700, found: 390.1709.

**(Z)-6-hydroxy-2-((5-methoxybenzofuran-2-yl)methylene)-7-((3-methylpiperidin-1-yl)methyl)benzofuran-3(2H)-one (8).** Prepared according to general procedure started with 0.67 mmol 6-hydroxy-3-coumaranonwith to afford the title compound (35 mg, 20%, white solid for step 1 using 3-methylpiperidine, and 12 mg, 52%, yellow solid for step 2 started with 0.015 mmol modified coumaranone derived from step 1 and 5-methoxybenzofuran-2-carbaldehyde); ^1^H NMR (400 MHz, Chloroform-*d*) δ 7.64 (dd, *J* = 8.4, 2.2 Hz, 1H), 7.45 (dd, *J* = 8.9, 2.2 Hz, 1H), 7.31 (d, *J* = 2.3 Hz, 1H), 7.13 (d, *J* = 2.5 Hz, 1H), 7.01 (dt, *J* = 9.0, 2.5 Hz, 1H), 6.88 (d, *J* = 2.2 Hz, 1H), 6.68 (dd, *J* = 8.5, 2.2 Hz, 1H), 4.05 (d, *J* = 2.2 Hz, 2H), 3.91 (s, 3H), 3.24–2.98 (m, 2H), 2.32 (br, 1H), 2.03 (br, 1H), 1.90–1.82 (m, 3H), 1.79–1.68 (m, 1H), 1.07 (br, 1H), 0.99 (d, *J* = 5.4 Hz, 3H); ^13^C NMR (100 MHz, Chloroform-*d*) δ 181.9, 168.7, 165.6, 156.4, 151.4, 150.8, 148.3, 129.5, 125.3, 115.6, 114.1, 113.2, 112.2, 111.8, 104.3, 103.2, 100.1, 77.4, 61.1, 56.0, 54.3, 53.7, 32.3, 31.4, 19.5; HRMS (FAB) *m*/*z* calcd. for C_25_H_26_NO_5_^+^ [M]^+^: 420.1805, found: 420.1809.

**(Z)-2-(benzofuran-2-ylmethylene)-6-hydroxy-7-(piperazin-1-ylmethyl)benzofuran-3(2H)-one (9).** Prepared according to following steps: step 1 started with 0.67 mmol 6-hydroxy-3-coumaranon and tert-butyl piperazine-1-carboxylate to afford a desired intermediate (100 mg, 43%, white solid), and step 2 started with the 0.15 mmol modified coumaranone derived from step 1 and benzofuran-2-carbaldehyde to afford a corresponding aurone derivative (40 mg, 58%, yellow solid). 0.063 mmol of desired product of step 2 was used for a next reaction with 2.8 mL of 4 M HCl in dioxane solution and 2.8 mL dichloromethane as a solvent. Reaction mixture was stirred for 2 h at room temperature. Precipitated yellow solid was collected by filtration, washed with water, and dried in vacuum to afford the tittle compound (23 mg, 97%, yellow solid); ^1^H NMR (400 MHz, DMSO-*d_6_*) δ 9.82 (s, 2H), 7.85 (d, *J* = 7.7 Hz, 1H), 7.80–7.74 (m, 2H), 7.46–7.38 (m, 1H), 7.32 (t, *J* = 7.5 Hz, 1H), 7.09 (d, *J* = 8.6 Hz, 1H), 6.94 (s, 1H), 4.47 (s, 2H), 3.61 (br, 4H), 3.43 (br, 4H); ^13^C NMR (100 MHz, DMSO-*d_6_*) δ 180.7, 167.1, 166.2, 155.1, 151.3, 149.6, 147.1, 128.5, 127.3, 126.6, 123.6, 122.4, 113.3, 112.9, 112.8, 111.6, 99.8, 47.8, 47.5, 40.3; HRMS (EI) *m*/*z* calcd. for C_22_H_20_N_2_O_4_^+^ [M]^+^: 376.1423, found:376.1421.

**(Z)-6-hydroxy-2-((5-methoxybenzofuran-2-yl)methylene)-7-(piperazin-1-ylmethyl)benzofuran-3(2H)-one (10).** Prepared according to following steps: step 1 started with 0.67 mmol 6-hydroxy-3-coumaranon and tert-butyl piperazine-1-carboxylate to afford a desired intermediate (100 mg, 43%, white solid), and step 2 started with the 0.15 mmol modified coumaranone derived from step 1 and 5-methoxybenzofuran-2-carbaldehyde to afford a corresponding aurone derivative (35 mg, 69%, yellow solid). 0.039 mmol of desired product of step 2 was used for a next reaction with 0.043 mL 1 M BBr_3_ in dichloromethane solution with 1 mL of dichloromethane solvent. The mixture was stirred for 50 h at 0 °C which was changed to room temperature slowly. After reaction followed by basic extractive work up, crude was purified with recrystallization using methylene chloride and hexane to afford the title compound (13 mg, 48%, orange solid); ^1^H NMR (400 MHz, DMSO-*d_6_*) δ 7.53 (d, *J* = 8.9 Hz, 1H), 7.40–7.34 (m, 2H), 7.25 (d, *J* = 2.6 Hz, 1H), 6.98 (dd, *J* = 9.0, 2.7 Hz, 1H), 6.58 (s, 1H), 6.38 (d, *J* = 8.6 Hz, 1H), 3.85 (s, 2H), 3.82 (s, 3H), 2.98 (br, *J* = 5.0 Hz, 4H), 2.74 (br, *J* = 5.0 Hz, 4H); ^13^C NMR (126 MHz, DMSO-*d_6_*) δ 181.7, 167.0, 155.9, 151.5, 149.5, 129.3, 124.5, 117.1, 116.4, 114.8, 111.7, 110.4, 105.2, 103.4, 98.9, 96.0, 83.2, 55.6, 50.4, 49.9, 43.8; HRMS (FAB) *m*/*z* calcd. for C_23_H_23_N_2_O_5_^+^ [M]^+^: 407.1601, found: 407.1611.

**(Z)-6-hydroxy-2-((5-hydroxybenzofuran-2-yl)methylene)-7-(piperazin-1-ylmethyl)benzofuran-3(2H)-one (11).** Prepared according to following steps: step 1 started with 0.67 mmol 6-hydroxy-3-coumaranon and tert-butyl piperazine-1-carboxylate to afford a desired intermediate (100 mg, 43%, white solid), and step 2 started with the 0.15 mmol modified coumaranone derived from step 1 and 5-methoxybenzofuran-2-carbaldehyde to afford a corresponding aurone derivative (35 mg, 49%, yellow solid). 0.069 mmol of desired product of step 2 was used for a next reaction with 0.69 mL 1 M BBr_3_ in dichloromethane solution with 1 mL of dichloromethane solvent. The mixture was stirred for 24 h at 0 °C which was changed to room temperature slowly. After reaction followed by basic extractive work up, crude was purified with flash chromatography on silica gel (dichloromethane/0.05 M ammonia in methanol = 10: 1) to afford the title compound (13 mg, 48%, orange solid); ^1^H NMR (400 MHz, DMSO-*d_6_*) δ 7.40 (d, *J* = 8.8 Hz, 1H), 7.33–7.25 (m, 2H), 7.00 (d, *J* = 2.5 Hz, 1H), 6.82 (dd, *J* = 8.8, 2.5 Hz, 1H), 6.50 (s, 1H), 6.30 (d, *J* = 8.6 Hz, 1H), 3.82 (s, 2H), 3.17 (s, 1H), 2.94 (t, *J* = 4.9 Hz, 4H), 2.70 (t, *J* = 4.9 Hz, 4H); ^13^C NMR (100 MHz, DMSO-*d_6_*) δ 178.0, 167.6, 154.1, 151.8, 150.3, 149.3, 129.9, 124.9, 122.5, 117.4, 115.3, 111.8, 110.3, 107.3, 105.9, 105.6, 96.2, 51.2, 50.4, 44.4; HRMS (EI) *m*/*z* calcd. for C_22_H_20_N_2_O_5_^+^ [M]^+^: 392.1372, found: 392.1373.

#### 3.4.5. Synthesis of Compound E

Triethylamine (0.841 mL, 6.03 mmol) and 3-picolylamine (0.563 mL, 5.53 mmol) were added to a solution of 2,4-dichloroquinazoline (1.0 g, 5.03 mmol) in tetrahydrofuran (13 mL). The mixture was stirred at room temperature. After completion of conversion, the reaction mixture was concentrated in vacuo, dissolved with CH_2_Cl_2_ (25 mL) and washed with distilled water (10 mL × 3). The organic layer was dried over MgSO_4_ and the solvent was evaporated. Resulting residue was purified by filtration washed by *n*-hexanes to obtain the intermediate.

The mixture of aforementioned intermediate (0.185 mmol), 2-hydroxyphenylboronic acid (30 mg, 0.222 mmol), Pd(PPh_3_)_4_ (21.3 mg, 0.019 mmol), and 2 M aqueous Na_2_CO_3_ (0.185 mL, 0.370 mmol) in 1,2-dimethoxyethane (0.5 mL) was added in a 10 mL microwave vial and heated at 150 °C under microwave irradiation. After 1 h, distilled water (10 mL) was added to the cooled reaction mixture and extracted with CH_2_Cl_2_ (25 mL × 3) and washed with brine (50 mL). The organic layer was dried over MgSO_4_ and the solvent was evaporated. The residue was purified by flash chromatography on silica gel (CH_2_Cl_2_/MeOH = 40:1) to give the desired product compound.

**2-(4-((3-(diethylamino)propyl)amino)quinazolin-2-yl)phenol (12).** Prepared according to general procedure, 2,4-dichloroquinazoline started to afford the title compound (51 mg, 85%, pale yellow solid); ^1^H NMR (400 MHz, DMSO-*d*_6_) δ 14.77 (s, 1H), 8.82 (t, *J* = 5.3 Hz, 1H), 8.47 (dd, *J* = 8.2, 1.8 Hz, 1H), 8.26 (dd, *J* = 8.3, 1.3 Hz, 1H), 7.80 (ddd, *J* = 8.3, 6.8, 1.3 Hz, 1H), 7.74 (dd, *J* = 8.3, 1.3 Hz, 1H), 7.53 (ddd, *J* = 8.2, 6.8, 1.4 Hz, 1H), 7.36 (ddd, *J* = 8.2, 7.2, 1.8 Hz, 1H), 6.95–6.84 (m, 2H), 3.69 (q, *J* = 6.6 Hz, 2H), 2.72–2.52 (m, 6H), 2.00–1.77 (m, 2H), 0.99 (t, *J* = 7.1 Hz, 6H). ^13^C NMR (101 MHz, DMSO-*d*_6_) δ 160.9, 160.8, 159.0, 146.7, 133.4, 132.4, 129.0, 126.1, 125.8, 122.9, 119.2, 118.1, 117.2, 113.5, 50.1, 46.3, 25.3, 11.2.

**3-(4-((3-(diethylamino)propyl)amino)quinazolin-2-yl)-4-hydroxybenzonitrile (13).** Prepared according to general procedure, 2,4-dichloroquinazoline started to afford the title compound (5 mg, 8%, pale yellow solid); ^1^H NMR (400 MHz, DMSO-*d*_6_) δ 16.07 (s, 1H), 9.02 (s, 1H), 8.79 (d, *J* = 2.3 Hz, 1H), 8.32 (d, *J* = 8.2 Hz, 1H), 7.89–7.82 (m, 2H), 7.79 (dd, *J* = 8.6, 2.2 Hz, 1H), 7.60 (ddd, *J* = 8.2, 6.6, 1.5 Hz, 1H), 7.09 (d, *J* = 8.6 Hz, 1H), 3.76 (d, *J* = 6.5 Hz, 2H), 2.75 (br, 6H), 1.94 (d, *J* = 25.9 Hz, 2H), 1.08 (s, 6H). ^13^C NMR (126 MHz, DMSO-*d*_6_) δ 165.4, 159.6, 159.6, 146.1, 136.1, 134.3, 133.9, 127.0, 126.3, 123.5, 120.1, 119.8, 119.5, 114.1, 100.8, 46.8.

**2-(4-((thiazol-4-ylmethyl)amino)quinazolin-2-yl)phenol (14).** Prepared according to general procedure, 2,4-dichloroquinazoline started to afford the title compound (42 mg, 70%, pale yellow solid); ^1^H NMR (400 MHz, DMSO-*d*_6_) δ 14.66 (s, 1H), 9.32 (s, 1H), 9.08 (s, 1H), 8.38 (dd, *J* = 8.4, 4.1 Hz, 2H), 7.83 (t, *J* = 7.6 Hz, 1H), 7.77 (d, *J* = 8.2 Hz, 1H), 7.60 (s, 1H), 7.54 (d, *J* = 7.6 Hz, 1H), 7.34 (t, *J* = 7.6 Hz, 1H), 6.89 (t, *J* = 6.7 Hz, 2H), 5.05 (d, *J* = 5.6 Hz, 2H). ^13^C NMR (151 MHz, DMSO-*d*_6_) δ 160.8, 160.7, 159.0, 152.0, 146.7, 137.7, 136.3, 133.6, 132.4, 129.1, 126.1, 125.9, 123.1, 119.1, 118.3, 117.2, 113.5, 36.7.

**4-hydroxy-3-(4-((thiazol-4-ylmethyl)amino)quinazolin-2-yl)benzonitrile (15).** Prepared according to general procedure, 2,4-dichloroquinazoline started to afford the title compound (39 mg, 60%, pale yellow solid); ^1^H NMR (400 MHz, DMSO-d_6_) δ 15.83 (s, 1H), 9.45 (t, *J* = 5.6 Hz, 1H), 9.09 (d, *J* = 2.0 Hz, 1H), 8.66 (d, *J* = 2.3 Hz, 1H), 8.38 (d, *J* = 8.2 Hz, 1H), 7.88–7.82 (m, 1H), 7.82–7.78 (m, 1H), 7.74 (dd, *J* = 8.5, 2.3 Hz, 1H), 7.61 (d, *J* = 1.8 Hz, 1H), 7.60–7.56 (m, 1H), 7.04 (d, *J* = 8.6 Hz, 1H), 5.06 (d, *J* = 5.6 Hz, 2H). ^13^C NMR (101 MHz, DMSO-d_6_) δ 164.7, 159.2, 159.1, 154.3, 154.1, 145.9, 135.6, 133.9, 133.7, 126.6, 125.9, 123.2, 119.7, 119.3, 118.9, 115.5, 113.7, 100.5, 41.0.

**3-(4-(((1H-pyrrol-3-yl)methyl)amino)quinazolin-2-yl)-4-hydroxybenzonitrile (16).** Prepared according to general procedure, 2,4-dichloroquinazoline started to afford the title compound (18 mg, 27%, yellow solid); ^1^H NMR (400 MHz, DMSO-d_6_) δ 16.08 (s, 1H), 10.65 (s, 1H), 9.17 (t, *J* = 5.6 Hz, 1H), 8.81 (t, *J* = 1.9 Hz, 1H), 8.36 (d, *J* = 8.3 Hz, 1H), 7.84 (d, *J* = 7.9 Hz, 1H), 7.79 (s, 1H), 7.77 (s, 1H), 7.55 (t, *J* = 7.6 Hz, 1H), 7.08 (d, *J* = 8.5 Hz, 1H), 6.82 (s, 1H), 6.71–6.57 (m, 1H), 6.13 (s, 1H), 4.74 (d, *J* = 5.6 Hz, 2H). ^13^C NMR (101 MHz, DMSO) δ 165.0, 159.0, 158.8, 145.9, 135.6, 133.8, 133.5, 126.4, 125.9, 123.2, 119.8, 119.6, 119.5, 119.1, 117.8, 116.2, 113.8, 107.7, 100.4, 38.9.

**4-hydroxy-3-(4-((oxazol-4-ylmethyl)amino)quinazolin-2-yl)benzonitrile (17).** Prepared according to general procedure, 2,4-dichloroquinazoline started to afford the title compound (58 mg, 88%, pale yellow solid); ^1^H NMR (600 MHz, DMSO-d_6_) δ 15.85 (s, 1H), 9.28 (t, *J* = 5.4 Hz, 1H), 8.71 (d, *J* = 2.2 Hz, 1H), 8.36 (s, 1H), 8.34 (d, *J* = 8.2 Hz, 1H), 8.07 (s, 1H), 7.84 (t, *J* = 7.7 Hz, 1H), 7.78 (d, *J* = 8.3 Hz, 1H), 7.75 (dd, *J* = 8.5, 2.2 Hz, 1H), 7.57 (t, *J* = 7.7 Hz, 1H), 7.04 (d, *J* = 8.5 Hz, 1H), 4.82 (d, *J* = 5.4 Hz, 2H). ^13^C NMR (151 MHz, DMSO-d_6_) δ 164.7, 159.1, 159.0, 152.1, 145.8, 137.1, 136.3, 135.6, 133.9, 133.6, 126.5, 125.9, 123.2, 119.6, 119.3, 118.7, 113.6, 100.5, 36.8.

**3-(4-(((1H-imidazol-4-yl)methyl)amino)quinazolin-2-yl)-4-hydroxybenzonitrile (18).** Prepared according to general procedure, 2,4-dichloroquinazoline started to afford the title compound (20 mg, 30%, pale yellow solid); ^1^H NMR (400 MHz, DMSO-*d*_6_) δ 15.98 (s, 1H), 11.92 (s, 1H), 9.25 (s, 1H), 8.78 (s, 1H), 8.38 (d, *J* = 8.3 Hz, 1H), 7.79 (dd, *J* = 21.6, 8.4 Hz, 3H), 7.58 (d, *J* = 15.2 Hz, 2H), 7.06 (d, *J* = 8.7 Hz, 2H), 4.81 (s, 2H). ^13^C NMR (101 MHz, DMSO-*d*_6_) δ 165.4, 159.5, 159.4, 146.3, 136.1, 135.4, 134.3, 134.1, 126.9, 126.3, 123.7, 120.2, 119.9, 119.5, 114.2, 113.9, 100.9.

**4-hydroxy-3-(4-(((1-methyl-1H-imidazol-4-yl)methyl)amino)quinazolin-2-yl)benzonitrile (19).** Prepared according to general procedure, 2,4-dichloroquinazoline started to afford the title compound (38 mg, 60%, pale yellow solid); ^1^H NMR (600 MHz, DMSO-d_6_) δ 15.96 (s, 1H), 9.26 (t, *J* = 5.6 Hz, 1H), 8.78 (d, *J* = 2.1 Hz, 1H), 8.38 (d, *J* = 8.3 Hz, 1H), 7.84 (t, *J* = 7.6 Hz, 1H), 7.80 (d, *J* = 8.2 Hz, 1H), 7.78 (dd, *J* = 8.7, 2.0 Hz, 1H), 7.57 (t, *J* = 7.6 Hz, 1H), 7.52 (s, 1H), 7.08 (s, 1H), 7.06 (d, *J* = 2.5 Hz, 1H), 4.78 (d, *J* = 5.6 Hz, 2H), 3.59 (s, 3H).^13^C NMR (151 MHz, DMSO-d_6_) δ 164.9, 159.1, 158.9, 145.9, 138.6, 137.4, 135.6, 133.8, 133.7, 126.5, 125.9, 123.2, 119.8, 119.4, 118.9, 117.8, 113.7, 100.4, 38.9, 32.8. 

**3-(4-((2-(1H-imidazol-4-yl)ethyl)amino)quinazolin-2-yl)-4-hydroxybenzonitrile (20).** Prepared according to general procedure, 2,4-dichloroquinazoline started to afford the title compound (41 mg, 63%, pale yellow solid); ^1^H NMR (400 MHz, DMSO-*d*_6_) δ 16.00 (s, 1H), 11.87 (s, 1H), 8.97 (s, 1H), 8.81 (d, *J* = 2.3 Hz, 1H), 8.27 (d, *J* = 8.2 Hz, 1H), 7.88–7.79 (m, 1H), 7.78–7.74 (m, 2H), 7.60–7.52 (m, 2H), 7.05 (d, *J* = 8.6 Hz, 1H), 6.88 (s, 1H), 3.89 (d, *J* = 6.6 Hz, 2H), 2.98 (t, *J* = 7.4 Hz, 2H). ^13^C NMR (101 MHz, DMSO-*d*_6_) δ 165.3, 159.5, 159.5, 146.2, 136.0, 135.3, 134.2, 134.1, 126.9, 126.3, 123.4, 120.1, 119.8, 119.3, 114.1, 100.8, 41.8, 40.6.

**4-hydroxy-3-(4-(((1-methyl-1H-imidazol-4-yl)methyl)amino)quinazolin-2-yl)benzaldehyde (21).** Prepared according to general procedure, 2,4-dichloroquinazoline started to afford the title compound (12 mg, 42%, pale yellow solid); ^1^H NMR (400 MHz, DMSO-*d*_6_) δ 16.06 (s, 1H), 9.94 (s, 1H), 9.26 (s, 1H), 9.02 (d, *J* = 2.2 Hz, 1H), 8.39 (d, *J* = 8.3 Hz, 1H), 7.90 (dd, *J* = 8.5, 2.2 Hz, 1H), 7.89–7.76 (m, 2H), 7.56 (ddd, *J* = 8.1, 6.5, 1.6 Hz, 1H), 7.52 (s, 1H), 7.14 (s, 1H), 7.09 (d, *J* = 8.5 Hz, 1H), 4.78 (d, *J* = 4.9 Hz, 2H), 3.58 (s, 3H). ^13^C NMR (126 MHz, DMSO-*d*_6_) δ 191.6, 167.1, 160.2, 159.3, 146.4, 134.2, 133.3, 132.9, 128.0, 126.8, 126.2, 123.7, 119.5, 118.9, 118.5, 114.1, 113.3, 99.9, 33.3.

### 3.5. Enzyme Inhibition Assays

The inhibitory activities of all compounds with respect to the wild type and the d746-750/T790M/C797S mutant of EGFR were measured by Reaction Biology Corp. (Malvern, PA, USA) using the radiometric kinase assays ([γ-^33^P]-ATP). Briefly, the enzymatic reaction mixtures contained an artificial substrate peptide, poly-Glu-Tyr (4:1) in freshly prepared base reaction buffer (20 mM HEPES of pH 7.5, 10 mM MgCl_2_, 1 mM EGTA, 0.02% Brij-35, 0.02 mg/mL BSA, 0.1 mM Na_3_VO_4_, 2 mM DTT, 1% DMSO). Wild-type and mutant EGFR were delivered into the substrate solution and gently mixed. Each inhibitor in 100% DMSO in a serial dilution was then delivered to the reaction mixture using the acoustic dispensing system (Echo550; nanoliter range). To initiate the enzymatic reaction, ^33^P-ATP with a specific activity of 10 μCi/μL was added into the reaction mixture, and the mixture was further incubated for 2 h at room temperature. Radioactivity was then monitored by the filter-binding method after reactions were spotted onto P81 ion exchange paper and the filters were washed extensively in 0.75% phosphoric acid. Kinase activity data were expressed as the percent remaining kinase activity in test samples compared to vehicle (dimethyl sulfoxide) reactions. IC_50_ values and curve fits were calculated with Prism Software (GraphPad Software) at 10 different concentrations. Staurosporine was employed as the positive control in this study.

### 3.6. Cell Proliferation Inhibition Assay

Two compounds (**18** and **19**) were sent to WuXi AppTec Corp., Shanghai, China, for cell proliferation inhibition assay. The cells were treated with various concentrations of the compounds in duplicate. After 72 h, the fluorescence signals were measured at an excitation wavelength of 540 nm and an emission wavelength of 590 nm using a microplate reader. More specifically, the number of living cells was determined with CellTiter-Glo^®^ Luminescent Cell Viability Assay (Promega).

## 4. Conclusions

Based on the two-track virtual screening and the targeted de novo design, we discovered the effective fourth-generation EGFR inhibitors highly selective for the d746-750/T790M/C797S mutant over the wild type. This was made possible by virtue of the modified protein–ligand binding free energy function involving a new hydration free energy term with enhanced accuracy. Most remarkably, compounds **19** and **20** exhibited low-nanomolar biochemical potency against EGFR^d746-750/T790M/C797S^ as well as more than 10^4^-fold selectivity over the wild type in vitro. The docking simulation results indicated that these new inhibitors would be bound tightly in the ATP-binding pocket of the triple mutant through the bidentate hydrogen bonds with backbone groups in the hinge region, together with the hydrophobic interactions with the nonpolar residues in the Gly loop, hinge region, and interdomain region. A hydrogen bond with the mutated residue Ser797 was also shown to be a significant binding force for the fourth-generation EFGR inhibitors to be stabilized in the ATP-binding site of the d746-750/T790M/C797S mutant. In addition to the additional interactions with the mutated residue Met790, the establishment of a van der Waals contact with Gly loop residues and the formation of a hydrogen bond with Asp855 in the activation loop could be invoked to account for the high inhibitory activity and selectivity of **19** and **20**, respectively. These fourth-generation EGFR inhibitors are anticipated to serve as a lead compound for the development of new anticancer medicines against NSCLC cells resistant to second- and third-generation EGFR inhibitor drugs.

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
