# Peer review of "Rational Computational Design of Fourth-Generation EGFR Inhibitors to Combat Drug-Resistant Non-Small Cell Lung Cancer"

_ijms, 2020, doi:10.3390/ijms21239323_

Round 1

Reviewer 1 Report

Park et al. describe virtual screening and de novo design and selection of fourth-generation EGFR inhibitors for EGFR+ NSCLC. The report is well-written and the subject of major clinical interest. I have some minor comments:
1. Introduction, first paragraph: EGFR mutations are driving only approximately 10-15% of lung adenocarcinomas, not all NSCLC; please rephrase.
2. Abstract: it lacks the information, how many compounds (potential inhibitors) were found at the end, and how good they are. Also it lacks the information about subsequent chemical synthesis/modifications.

Author Response

1) Introduction, first paragraph: EGFR mutations are driving only approximately 10-15% of lung adenocarcinomas, not all NSCLC; please rephrase.

Following the suggestion, we have indicated that EGFR mutations are responsible for the pathogenesis and the progression of approximately 10-15% of lung adenocarcinomas on p. 3 line 3 in the revised manuscript.

2) Abstract: it lacks the information, how many compounds (potential inhibitors) were found at the end, and how good they are. Also it lacks the information about subsequent chemical synthesis/modifications.

In Abstract section of the revised manuscript, we have summarized the results of virtual screening and chemical modifications to find the fourth-generation EGFR inhibitors in concomitance with the merits of the representative compounds.

Reviewer 2 Report

The manuscript entitled "Rational Computational Design of Fourth-Generation
EGFR Inhibitors to Combat Drug-Resistant Non-Small Cell Lung Cancer" highlighted the role of fourth-generation EGFR inhibitors to selectively inhibit the d746-750/T790M/C797S mutant over the wild type.

The manuscript is really well written and of interest for the audience.

Minor concerns:

  • Gene acronyms should be written in italics.
  • Mutations should be reported as follow: EGFR exon 20 p.T790M/EGFR exon 19 p.E746_A750del.

Author Response

In accordance with the comments, we have corrected the typos in gene acronyms on p. 22 and the notation of mutant on p. 2 in the revised manuscript.

This manuscript is a resubmission of an earlier submission. The following is a list of the peer review reports and author responses from that submission.